# Relationships between Springtime PM$_{2.5}$, PM$_{10}$, and O$_3$ Pollution and the Boundary Layer Structure in Beijing, China

**Qing Zhou** [1], **Lei Cheng** [2,*], **Yong Zhang** [2,*], **Zhe Wang** [2] and **Shili Yang** [3]

1   Meteorological Observation Center, China Meteorological Administration (CMA), Beijing 100081, China; zhouqing@cma.gov.cn
2   Institute for Development and Programme Design, China Meteorological Administration (CMA), Beijing 100081, China; wangzhe@cma.gov.cn
3   Beijing Meteorological Observation Centre, Beijing Meteorological Bureau, Beijing 100089, China; yangsl@mail.bnu.edu.cn
*   Correspondence: clei@cma.gov.cn (L.C.); yzhang@cma.gov.cn (Y.Z.)

**Abstract:** Complex pollution with high aerosol and ozone concentrations has recently been occurring in several densely populated cities in China, raising concerns about the influence of meteorological factors, including synoptic circulation and local conditions. In this study, comprehensive analyses on the associations between PM$_{2.5}$, PM$_{10}$, and O$_3$ and meteorological conditions were conducted based on observations from radar wind profiler, microwave radiometer, automatic weather station, and air quality monitoring sites in Beijing during the spring of 2019. The results showed that the boundary layer height and temperature inversion were negatively (positively) correlated with PM (O$_3$) concentrations, modulating the degree of air pollution. Five identified synoptic patterns were derived using geopotential height data of the ERA5 reanalysis, among which Type 1, characterised by south-westerly prevailing winds with high pressure to the south, was considered to be associated with severe PM and O$_3$ contamination. This indicates that air pollutants originating from southern regions exert a major influence on Beijing through the transportation effect. In addition, high temperature, relative humidity, and low wind velocity exacerbate pollution. Overall, this study provides significant information for understanding the vital roles played by meteorological elements at both the regional and local scales in regulating air contamination during spring in Beijing.

**Keywords:** PM and O$_3$ pollution; planetary boundary layer; synoptic patterns; springtime; Beijing

## 1. Introduction

With the increasing economic development and urbanisation in China [1], aerosol pollution and air quality issues have been attracting significant attention, and efforts have been made to determine the variations in the characteristics and causes of air pollution, including the influence of anthropogenic emissions [2,3] and meteorological factors [4–6]. As major air pollutants, PM$_{2.5}$ and tropospheric O$_3$ concentrations are coupled through various processes, including aerosol formation and growth, heterogeneous reactions, and photolysis induced by aerosol variation [7], thereby exerting significant influences on the weather, climate, environment, and human health [8,9]. PM$_{2.5}$ concentrations in China have considerably decreased over the past few years owing to the effective implementation of emission mitigation measures; however, O$_3$ concentrations have been increasing in most cities in China since 2013 [10,11]. Therefore, examining the relationships between O$_3$ and PM$_{2.5}$ is necessary to control these two pollutants. In addition, as a carrier and catalyst for various pollutants, PM$_{10}$ has a strong adsorption capacity and is associated with increased respiratory symptoms and other illnesses [12]. Thus, exploring the variations in the characteristics and improvement measures of PM$_{10}$ is imperative.

The extent of PM and O$_3$ pollution depends not only on population density, terrain, industrial layout, and the intensity of industrial sources but also on multiscale atmospheric

circulation patterns, which can modulate the formation, dispersion, and deposition of contaminants [4,13,14]. The East Asian monsoon exerts a significant impact on annual seasonal pollutant variations throughout China [15,16]. In addition, changes in local meteorological conditions, such as temperature, humidity, wind, precipitation, temperature inversion (TI), cloud cover, radiation intensity, atmospheric boundary height (BLH), and visibility, may also serve as indispensable factors associated with the production and removal of pollutants from the atmosphere [17,18]. Therefore, research on the association between atmospheric pollution and meteorological elements can clarify pollution generation mechanisms and aid pollution forecasts and warnings.

As the most modernised and populous metropolitan area in China, Beijing has recently been experiencing severe complex pollution, and many studies have been conducted on variations in PM and $O_3$ pollutant characteristics and the influence of meteorological elements on seasonal pollutant concentration in Beijing [4,6,19,20]. Previous research has demonstrated that heavy aerosol pollution in fall and winter in Beijing is frequently linked to synoptic patterns with southerly prevailing winds or patterns with weak north-westerly prevailing winds and a strong elevated TI phenomenon [20]. In summertime, the meteorological conditions of pollution occurrences show synergistic contributions of high relative humidity (RH), weak winds, low BLHs, and stable atmospheric stratifications to contamination under the background of southerly/south-westerly prevailing winds [6]. In addition, most severe pollution events occur in spring, resulting from dust storms [21,22]. Nevertheless, the pollution scenario and its association with meteorological factors during the spring in Beijing have rarely been studied. In addition, as an important factor that determines the vertical diffusion and convective mixing of pollutants, the BLH retrieved from radiosonde observations is typically used to analyse the relationship between pollution and meteorological elements. Owing to the low temporal resolution, radiosonde measurements cannot be utilised to continuously retrieve the variations in BLH. The development of ground-based remote sensing equipment, such as radar wind profilers (RWPs) and microwave radiometers (MWRs), with high temporal resolution, provides possibilities for examining successive variations in BLH and their association with meteorological conditions.

This study employs a climatological method [23] to elucidate the relationships between $PM_{2.5}$, $PM_{10}$, $O_3$, and meteorological factors in the boundary layer structure in Beijing during the spring of 2019 based on the continuous observation of BLHs and TIs retrieved from RWP and MWR, respectively. The remainder of this manuscript is organised as follows. The data and methods used are introduced in Section 2, followed by an analysis of the relationships between aerosols, ozone pollution, and meteorological factors in the planetary boundary layer (PBL) structure under different synoptic patterns in Section 3. Finally, the main findings are summarised in Section 4.

## 2. Materials and Methods

### 2.1. Data

In this study, hourly observations of $PM_{2.5}$, $PM_{10}$, and $O_3$ concentrations in Beijing during spring (March, April, and May) 2019 were derived, based on data from six air quality monitoring sites (blue squares in Figure 1). We employed the micro-oscillating balance method and/or β-absorption method [24] to retrieve aerosol mass concentrations and the UV spectrophotometry method to obtain ozone concentrations. Ground-level hourly 2-m temperature, 2-m RH, and 10-m wind speed and direction were also obtained from an automatic weather station (AWS) situated at the Beijing site (39.98° N, 116.28° E, 46.9 m above sea level, black triangle in Figure 1) to understand the dominant impacts of the PBL structure. The Beijing site was located in Haidian park in the Beijing urban area, surrounded by green space, so was less influenced by traffic air pollution [25].

To elucidate the associations between the PBL and air contamination in Beijing during spring, observations from the RWP and MWR collected from the Beijing site (39.98° N, 116.28° E, 20.4 m above sea level, green square and red dot in Figure 1, respectively) were processed and analysed for comparison with pollution and meteorological measurements.

The technical specification of RWP and MWR at the Beijing site have been summarized in Tables 1 and 2.

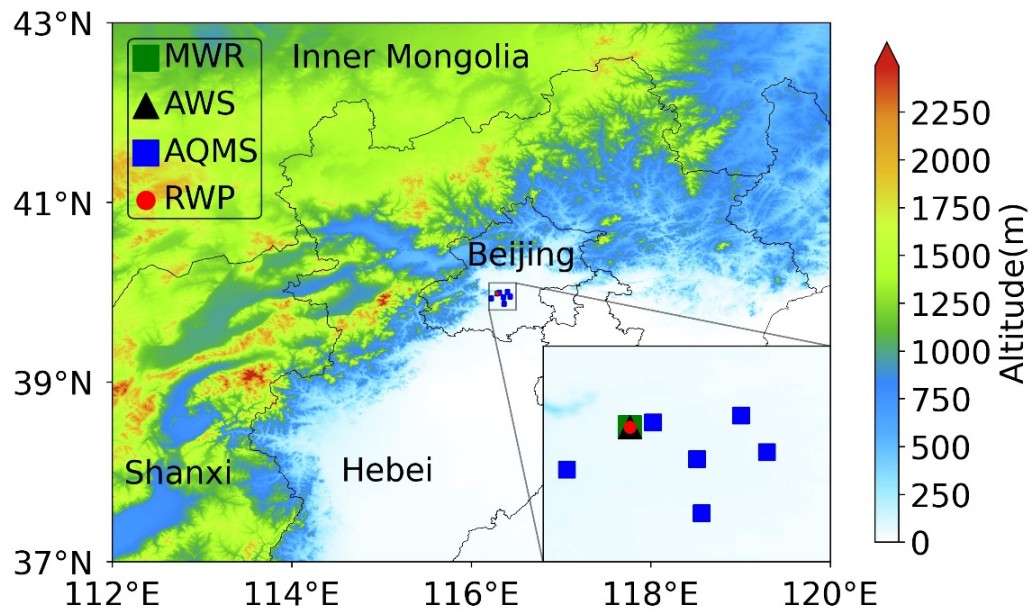

**Figure 1.** Terrain height map of northern China. The radar wind profiler (RWP) and microwave radiometer (MWR) stations (39.98° N, 116.28° E) in Beijing are denoted by a red dot and green square, respectively, and the automatic weather station (AWS) and air quality monitoring sites (AQMS) are marked by a black triangle and blue squares, respectively.

**Table 1.** Characteristics of the radar wind profiler (RWP) at the Beijing site.

| Parameter | Technical Specification |
| --- | --- |
| Frequency | 1297 MHz |
| Wavelength | 220 mm |
| Transmit peak power | 10 kW |
| Sampling period | 4.5 min |
| Pulse width | 12.8 μs (high model)/6.4 μs (moderate model)/0.8 μs (low model) |
| Vertical resolution | 120 m |
| Temporal resolution | 6 min |
| Sampling altitude range | 3150~10110 m (high model)/1110~4590 m (moderate model)/150~3630 m (low model) |

**Table 2.** Characteristics of the microwave radiometer (MWR) at the Beijing site.

| Parameter | Technical Specification |
| --- | --- |
| Frequency | 14 channels (smallest frequency of 22.24 GHz and largest frequency of 58.0 GHz) |
| Temporal resolution | 2 min |
| observation range of Bright temperature | 0~400 K |
| Accuracy of Bright temperature | 0.5 K |
| Detection altitude range | 0~10 km |
| vertical resolution (Temperature profile) | ≤50 m (0–500 m) ≤150 m (500–2000 m) ≤250 m (2000–10,000 m) |
| vertical resolution (Humidity profile) | ≤100 m (0–500 m) ≤200 m (500–2000 m) ≤400 m (2000–10,000 m) |

Observations from a RWP manufactured by the 23rd Institutes of China Aerospace Science and Industry Corporation were acquired during spring (March–May) 2019 at a time resolution of 6 min and a vertical resolution of 120 m. As one of the various products,

the refractive index structure constant ($C_n^2$) was used for the retrieval of BLHs. In particular, observations under clear-sky conditions were derived to calculate the BLHs to filter out cloud effects.

The multichannel MWR deployed in Beijing was a humidity and temperature microwave profiler (HTG4) manufactured by the Beijing Airda Electronic Equipment Corporation. It measured brightness temperatures (Tb) for fourteen channels (from 22.24 GHz to 58.0 GHz), with a temporal resolution of 2 min and Tb error $\leq$ 1 K. Atmospheric temperature and humidity profiles from the ground to an altitude of 10 km, as well as other products such as column-integrated water vapour and liquid water path, were retrieved from MWR observations using inversion algorithms. Based on the cloud height and thickness observed by Ka-band cloud radar, the quality control of brightness temperature was carried out, and a BP neural network model was established to realize the inversion of temperature and humidity profiles [26].

### 2.2. BLH Derived from RWP Measurements

The RWP detects the vertical wind field distribution by transmitting and receiving electromagnetic signals backscattered by atmospheric turbulence, which represents small-scale fluctuations in wind, temperature, and particularly humidity. In the entrainment layer of the ABL, where aerosol concentration and RH drastically change with altitude [27,28], the refractive index structure constant $C_n^2$ tends to reach a maximum at the BLH. A direct proportional relationship exists between $C_n^2$ and the signal-to-noise ratio (SNR) of the RWP [29], which provides the possibility of continuous BLH monitoring. As the selection of the SNR threshold correlates with local atmospheric conditions and RWP instrument performance, we adopted the method proposed by Liu et al. [30] to retrieve the BLH; a brief description of this method is provided below. First, the SNR profiles were normalised by dividing the original SNR profiles by the maximum SNR profiles (giving the NSNR). Subsequently, the threshold values ($\Delta$) were set as 0.75 in the daytime and 0.9 at night because of the diurnal variations in atmospheric turbulence, which is less active at night [31]. In the uncertainty analysis conducted by Liu et al. [30], where the inversion results of different thresholds were compared with those derived from radiosondes, the threshold values deployed had a good correlation and small deviation. Thus, the first junction point where the NSNR value intersected the threshold after the maximum NSNR value was identified as the BLH. Finally, the BLH results were derived using a multijunction filter to overcome the multiple local peaks caused by buoyancy fluctuations.

### 2.3. TIs Derived from the MWR

TI, which is considered as a capping layer overlying the PBL during the day, exerts a negative impact on atmospheric pollutant dispersion [32,33]. Based on the temperature profiles from the MWR described in Section 2.1, TIs were derived using the first-derivative algorithm [34,35]. First, the derivative at certain altitude intervals for each temperature profile was calculated. The derivative profiles were then scanned upward from the surface to 3000 m. An inversion layer was identified when the derivative was larger than zero and remained positive over an altitude range of more than 100 m. In addition, inversion layers with a temperature deviation of less than 0.5 °C between the top and base of the layer were discarded. For each inversion layer examined based on the above steps, the base height of the inversion (BH) and inversion intensity (°C/km) were calculated. The inversion intensity represents the temperature gradient, which is calculated as the temperature difference divided by the inversion depth ($\Delta T/\Delta Z$), where $\Delta Z$ is the height difference between the top and base of the inversion.

### 2.4. Classification of Synoptic Conditions

As a robust objective classification approach that can process large amounts of data with less subjective dependence [4,23,36], the obliquely rotated principal component analysis in the T-mode (T-PCA) approach was employed to sort the synoptic patterns related

to springtime aerosols and ozone pollution in Beijing by calculating the eigenvectors of geopotential height (GH) data to identify dominantly polluted synoptic conditions via singular value decomposition. Hourly 850-hPa GH fields obtained from ERA5 (the fifth generation ECMWF reanalysis for the global climate and weather, with a spatial resolution of $0.25° \times 0.25°$) (https://cds.climate.copernicus.eu/cdsapp#!/home) (accessed on 1 March 2019) in 2019 covering the study area (110–125° E, 35–45° N) were used, and five different synoptic patterns were identified based on the T-PCA method. The GH fields of rainy samples were discarded to eliminate the impact of wet scavenging.

### 2.5. Model Description and Configurations

To understand springtime pollution in Beijing, which can increase owing to the transportation of pollutants from potential source regions, we performed air mass trajectory analysis [22] to evaluate the transportation paths of contaminants to Beijing under various synoptic conditions. In this study, the hybrid single-particle Lagrangian integrated trajectory model (HYSPLIT) [37] developed by NOAA's Air Resources Laboratory was employed for its widespread application in contaminant transportation and diffusion [38]. Using HYSPLIT [39] and 6-h NCEP-FNL reanalysis [40], 24-h backward trajectories of each spring day during 2019 were retrieved. The NCEP FNL (Final) Operational Global Analysis data (NCEP-FNL) is produced by the Global Data Assimilation System, and continuously assimilates observations from the Global Telecommunication System and other sources. The NCEP-FNL reanalysis fields are on 1-degree by 1-degree grids, prepared operationally every six hours. The trajectory endpoint was set at the centre of Beijing (39.98° N, 116.28° E).

## 3. Results

### 3.1. Relationship between the PBL and Air Pollution in Beijing during Spring

During the spring of 2019, most BLHs in Beijing were less than 2.5 km. According to the diurnal variations in BLHs as shown in Figure 2, the BLHs showed a comparatively low value (average BLH = 0.42 km) during the period from 20:00 BJT to 05:00 BJT, began to increase after 08:00 BJT, and reached a peak value at 14:00 BJT (average BLH = 1.26 km). An anti-correlation was observed between the diurnal evolution of BLHs and $PM_{2.5}$ and $PM_{10}$ concentrations in Beijing, which peaked at night and reached a minimum between 14:00 BJT and 15:00 BJT (Figure 2a,b). However, the hourly correlation between the BLHs and $O_3$ concentration was generally positive, particularly from 10:00 to 15:00 BJT, during which $O_3$ pollution played an important role in Beijing owing to photochemical reactions.

To further illustrate the impact of BLH at varying aerosol and ozone concentrations, a kernel probability density estimation of BLHs as a function of $PM_{2.5}$, $PM_{10}$, and $O_3$ at different intervals is shown in Figure 3a–c, respectively. Kernel probability density (a violin plot) is a combination of a boxplot and a kernel density plot, where the boxplot shows the location of each quantile while the kernel density plot shows the density at any position. The larger the area of the violin, the greater the distribution probability of the sample value. As shown in the figure, the BLHs were both negatively correlated with $PM_{2.5}$ and $PM_{10}$ concentrations, with correlation coefficients of $-0.12$ and $-0.23$, respectively, illustrating that low BLHs limit the diffusion of pollutants and further aggravate pollution [41]. In contrast, a positive association was observed between the BLHs and $O_3$ concentration (R = 0.45), which is generally consistent with the results shown in Figure 2c. This was because solar radiation was strong during the daytime, when BLHs developed to a high altitude, and NOx and volatile organic compounds photochemically reacted in sunlight forming ozone.

In addition to BLH, another important PBL parameter related to the variations in aerosol and ozone pollution is TI, which deserves to be examined. As shown in Figure 4a, the BH of the TI was essentially anti-correlated with the $PM_{2.5}$ and $PM_{10}$ concentrations during the daytime in Beijing; however, a significant positive relationship was observed between the BH of the TI and $O_3$ concentration, with a correlation coefficient of 0.41, which

was analogous to the relationship between BLH and $O_3$ concentration (with a correlation coefficient of 0.45). The BH can also be considered as the height of the boundary layer because it covers the convective boundary layer during the daytime [42]. Figure 4b illustrates the functional relationship between the variations in $PM_{2.5}$, $PM_{10}$, and $O_3$ concentrations and the intensity. The intensity generally had a slightly positive association with $PM_{2.5}$, $PM_{10}$, and $O_3$ concentration, with a low correlation coefficient of less than 0.1.

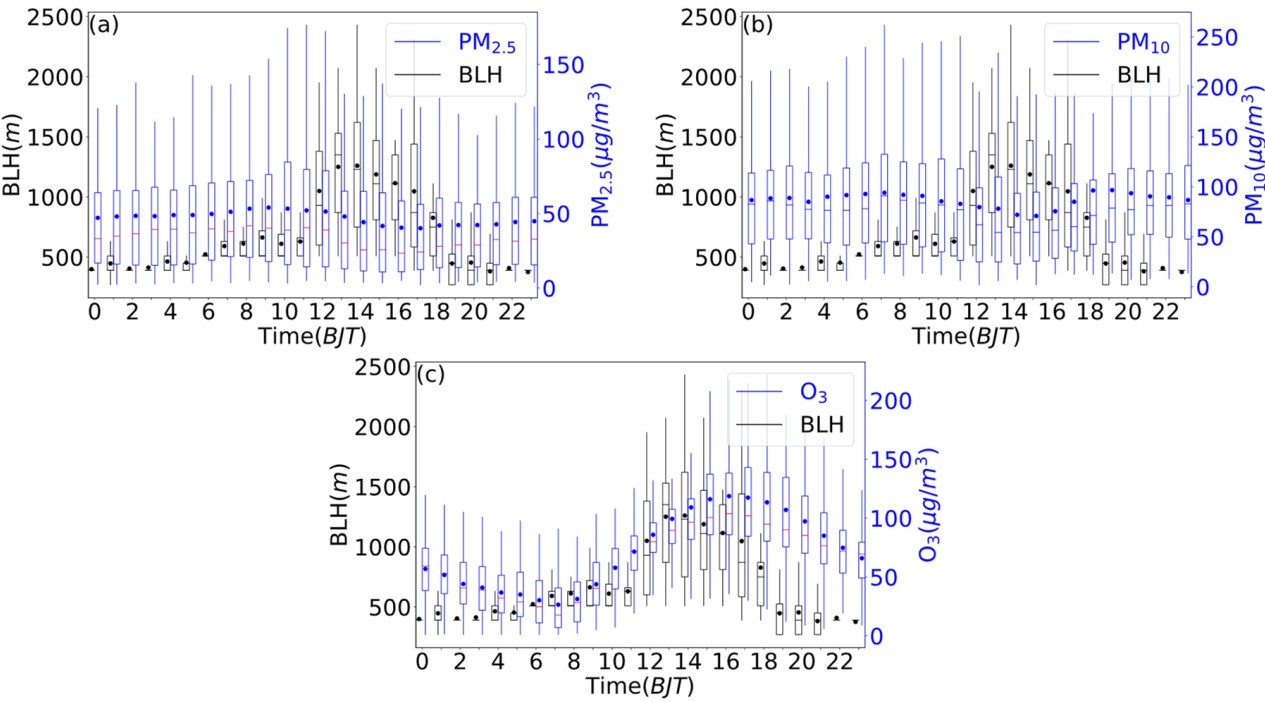

**Figure 2.** Box plot of diurnal variations in $PM_{2.5}$ (**a**,**b**), and $O_3$ concentrations (**c**) (in blue) and boundary layer height (BLH, in black) during spring in Beijing (representing the mean of the season). The central box represents the values from the 25th percentile to the 75th percentile; the vertical line extends from the lower to upper values (5th to 95th percentile); the middle solid line and dot denote the median and mean value, respectively.

### 3.2. Analysis of Synoptic Patterns Related to Air Pollution

Based on the 850-hPa GH field from the ERA-5 reanalysis described in Section 2.4, five major synoptic patterns were identified via the T-PCA method (Figure 5). Considering the spatial distribution of different pressure systems surrounding Beijing, each synoptic pattern can be described by the relative positions of the high-pressure and low-pressure systems as follows. Type 1: low pressure to the north, high pressure to the south; Type 2: low pressure to the northeast, high pressure to the southwest; Type 3: high pressure to the northwest and southeast, low pressure to the northeast; Type 4: high pressure to the east, low pressure to the west; and Type 5: high pressure to the northwest, low pressure to the west and east.

In these synoptic patterns, the $PM_{2.5}$ and $PM_{10}$ concentrations exhibited relatively high values in Type 1 (average of $66.1 \pm 42.4$ μg/m$^3$ and $113.5 \pm 61.5$ μg/m$^3$ for $PM_{2.5}$ and $PM_{10}$, respectively) and Type 3 (average of $61.3 \pm 47.4$ μg/m$^3$ and $118.7 \pm 82.5$ μg/m$^3$ for $PM_{2.5}$ and $PM_{10}$, respectively), which accounted for 38.3% and 9.3% of the total, respectively. As the second most frequent synoptic pattern in spring, Type 1 featured south-westerly prevailing winds driven by south-east to north-west pressure gradients, with a high-pressure system situated to the south-east of Beijing and a low-pressure system to the north (Figure 5). At the 850-hPa level, Type 4 were characterized by a high-pressure system located to east of Beijing, bringing southerly winds to Beijing, while Type 3 were characterized by a high-pressure system located to south-east and north-west of Beijing which experienced

weak northerly winds. Pollutants originating from southern regions in Hebei Province could be transmitted to Beijing during transportation by the southerly winds, resulting in air quality deterioration in Beijing. These results are consistent with those of previous studies that have revealed the influence of synoptic forcings on the transportation of contaminants to Beijing [4,14]. In contrast, the relatively clean Type 2 winds were related to north-westerly winds, with low pressure to the north-east and high pressure to the south-west (Figure 5), accounting for 44.7% of the total samples. In addition, under the dominance of Type 5, Beijing is situated at the bottom of the high-pressure ridge, with a moderate aerosol pollution level.

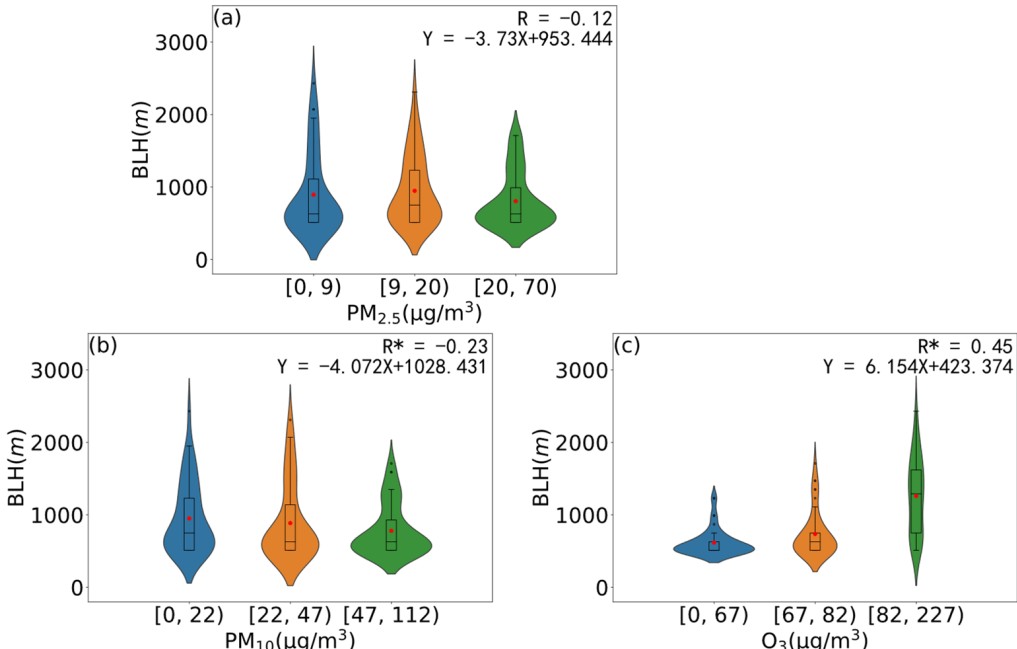

**Figure 3.** Kernel probability density estimates between BLH and (**a**) $PM_{2.5}$, (**b**) $PM_{10}$, and (**c**) $O_3$ concentrations during 08:00 BJT and 14:00 BJT in the study period. The samples in each bin are equal. the intervals of $PM_{2.5}$, $PM_{10}$ and $O_3$ concentrations are decided by the pollution concentration samples and each interval has the same amount of pollution concentrations samples according to the percentile of the sequence. Rectangles represent the maximum, minimum, median, and 50% range of the data in each interval. Red dots denote the average value in each interval. In the linear regression function, x represents $PM_{2.5}$ (**a**), $PM_{10}$ (**b**) and $O_3$ (**c**) concentration, respectively, and y represents BLH. The colour-filled areas denote the probability of samples, R represents the correlation coefficient between the average value in each interval, and BLH and R* indicate that the correlations are statistically significant ($p < 0.05$).

$O_3$ concentrations reached a maximum in Type 1 (with an average of $80.7 \pm 57.4$ μg/m³), which was located in front of the low-pressure area. The low pressure and relatively high temperature led to an increase in the rate of chemical reactions associated with ozone formation, leading to an increase in $O_3$ concentration. In addition, when the near surface was controlled by low pressure, the ambient pollutants rapidly converged to the centre, driven by the high-pressure air mass, resulting in a sharp increase in $O_3$ concentration near the low-pressure centre. In contrast, $O_3$ concentrations were relatively low for Type 5 when Beijing was dominated by high pressure. The air mass carrying pollutants in the centre of the high-pressure area rapidly diffused to the surrounding low-pressure area, which was conducive to the dilution and diffusion of ozone and its precursors, resulting in a low ozone concentration [19].

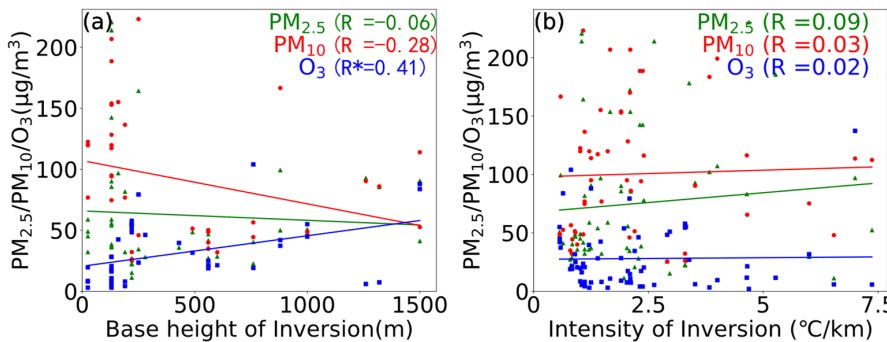

**Figure 4.** Scatter plots comparing PM$_{2.5}$ (green triangles), PM$_{10}$ (red boxes), and O$_3$ (blue squares) concentrations with (**a**) the base height (BH) and (**b**) inversion intensity during the period from 08:00 BJT to 14:00 BJT in Beijing during spring 2019. In the top right corner of each figure, the correlation coefficients (R) are shown, where the asterisks indicate that the correlations are statistically significant ($p < 0.05$).

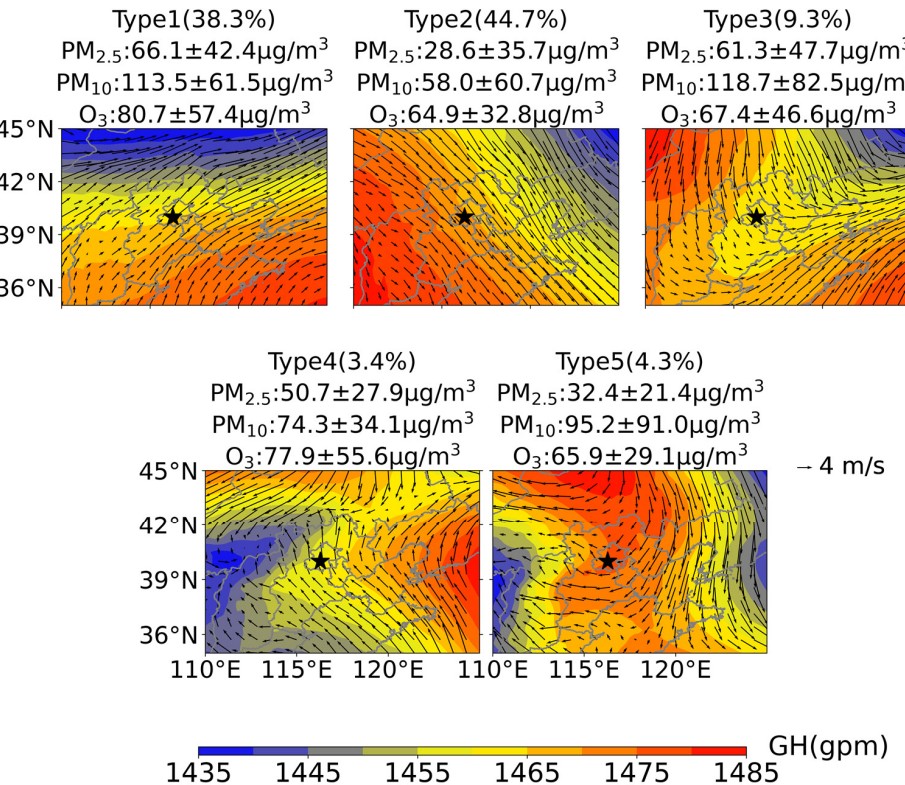

**Figure 5.** 850-hPa geopotential height (GH) (colour shading) superposed with the wind vector (arrows) in the circumstances of different springtime synoptic patterns in northern China. The occurrence frequency of each synoptic pattern and mean value and standard deviation of PM$_{2.5}$, PM$_{10}$, and O$_3$ are displayed at the top of each panel. The black star represents the location of Beijing.

Notably, one intriguing feature is that when the prevailing wind changed from north-west (Type 2) to south-west or south-east (Type 1 and Type 4), the pollution scenario in Beijing was exacerbated (Figure 5). To further elucidate the impact of dominant wind directions on pollution scenarios in Beijing, the observations of near-surface winds at different air pollution levels are illustrated in Figure 6. The top 30% of the PM and O$_3$ samples were characterized by south-westerly winds (Figure 6b,d,f). In contrast, the bottom 30% of the PM and O$_3$ samples were consistently related to north-westerly and north-easterly winds (Figure 6a,c,e). As analysed above, this discrepancy may be related

to the atmospheric transportation of air masses originating from regions with different air qualities.

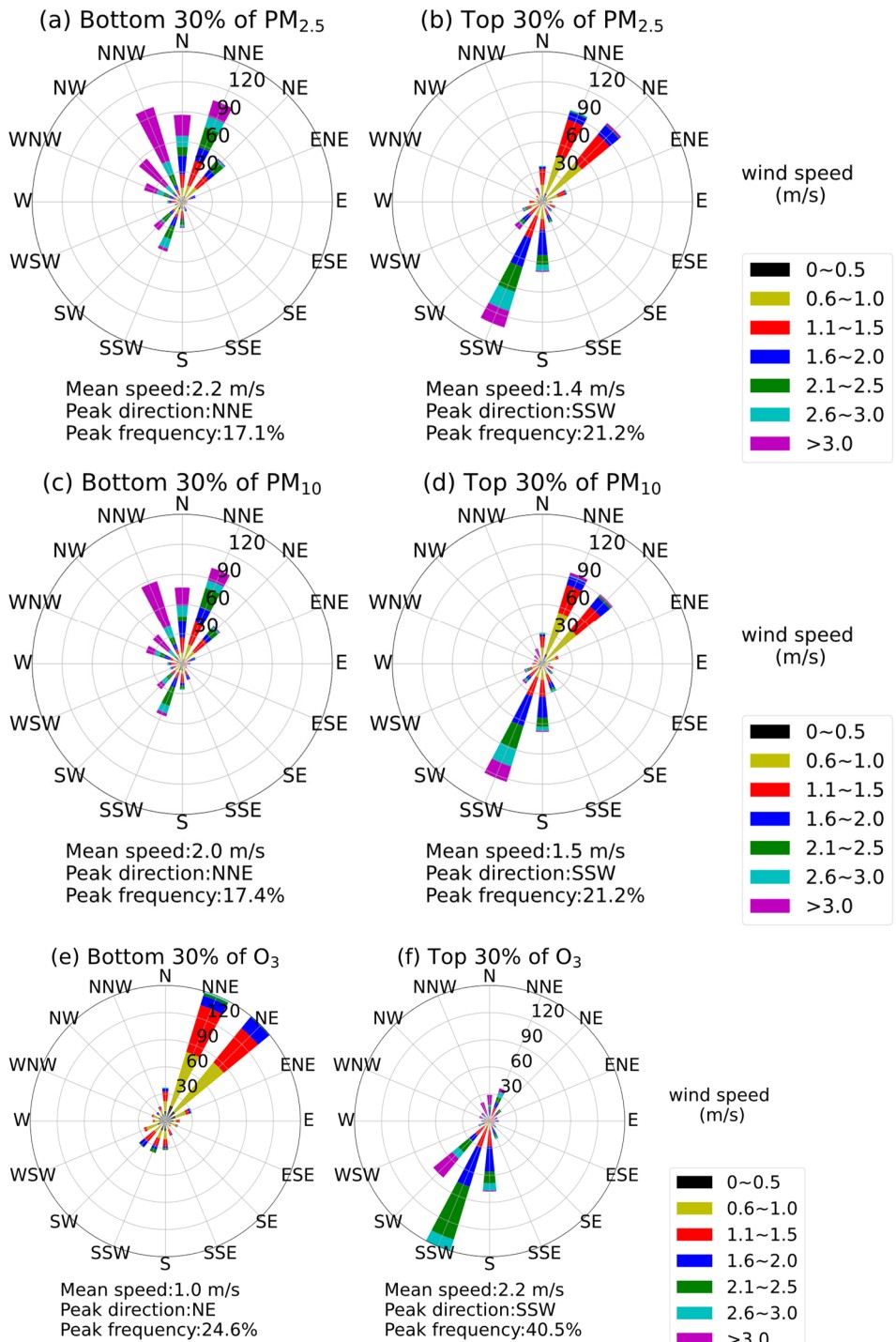

**Figure 6.** Wind rose diagrams showing the comparison of 10-m winds between the bottom 30% of PM$_{2.5}$ (**a**), PM$_{10}$ (**c**), and O$_3$ (**e**) concentrations and top 30% of PM$_{2.5}$ (**b**), PM$_{10}$ (**d**), and O$_3$ (**f**) concentrations in Beijing during spring 2019.

The 24-h backward trajectories under different synoptic types were calculated, presenting the potential contaminant transport routes (Figure 7). Air masses originating from north-western regions that had relatively lower emissions of PM and NOx than those in the Beijing District (Figure 8), were transmitted to Beijing under the influence of synoptic Type

2 [43]. Although Beijing is under control of north-western airmasses in Type 3, the pressure gradient is weak and wind speed is low, which is unfavorable to the diffusion of pollutants. However, affected by Types 1 and 4, Beijing was chiefly invaded by south-western and southern air masses, including those from southern Hebei and Shanxi provinces (Figure 7), which experienced heavy pollution in spring 2019 (Figure 8). Therefore, south-westerly prevailing winds under Types 1 and 4 exerted strong impacts on transmitting contamination from the southern regions to Beijing, deteriorating the aerosol and ozone pollution scenario. Under the impact of Type 5, Beijing was invaded by south-eastern air masses that originated in eastern Hebei provinces and Tianjin, with higher pollutant concentrations (Figure 8), leading to a moderate degree of air pollution.

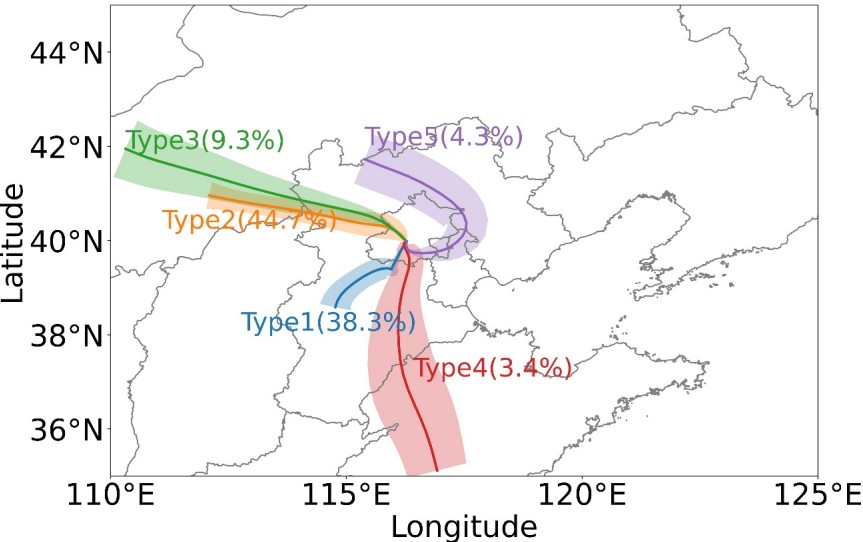

**Figure 7.** Cluster results of 24-h backward trajectories in the circumstances of five synoptic patterns. Solid curves and shaded areas represent the average trajectories and uncertainties, respectively. The occurrence frequency for each synoptic pattern is also displayed.

### 3.3. Meteorological Conditions Related to Synoptic Patterns

Local meteorological elements under the boundary layer could exert an impact on air pollution in Beijing, in addition to the influence of atmospheric transportation on air quality [4,23]. Figure 9 consists of box plots showing the observed aerosol concentration, $O_3$ concentration, and meteorological elements related with the different synoptic conditions. The background colour represents the pollution level in Beijing, which is defined as the PM concentration and $O_3$ concentration of each synoptic pattern. A severe pollution level represents high values of PM and $O_3$ concentration. As illustrated in Section 3.2, the aerosol and ozone concentration generally follows a downward trend from Type 1 to Type 2, that is, Type 1 > Type 3 > Type 4 > Type 5 > Type 2, which is coloured from reddish brown to green. Therefore, the pollution becomes severe as the colour gradually changes from green to reddish brown. As shown in Figure 9, Types 1, 3 and 4, which correspond to heavy aerosol pollution in Beijing, generally appeared on days with high temperatures, high RH at the surface, and low wind speeds. In comparison, the Type 2 pattern, which was associated with relatively light pollution, was characterized by lower temperature, humidity, and higher wind velocity. In the Type 5 pattern, when Beijing experienced a moderate air pollution level, the temperature and humidity were a bit higher than Type 2, while wind speed was a bit lower than Type 2. With an increase in wind velocity, BLHs were elevated, and vertical motion was continuously strengthened, thereby enhancing the horizontal diffusion capacity of the atmosphere and diluting aerosols and $O_3$ [44]. However, low wind velocity may lower BLHs, leading to the restriction of vertical mixing and dispersion of contaminants, thereby promoting the occurrence of pollution.

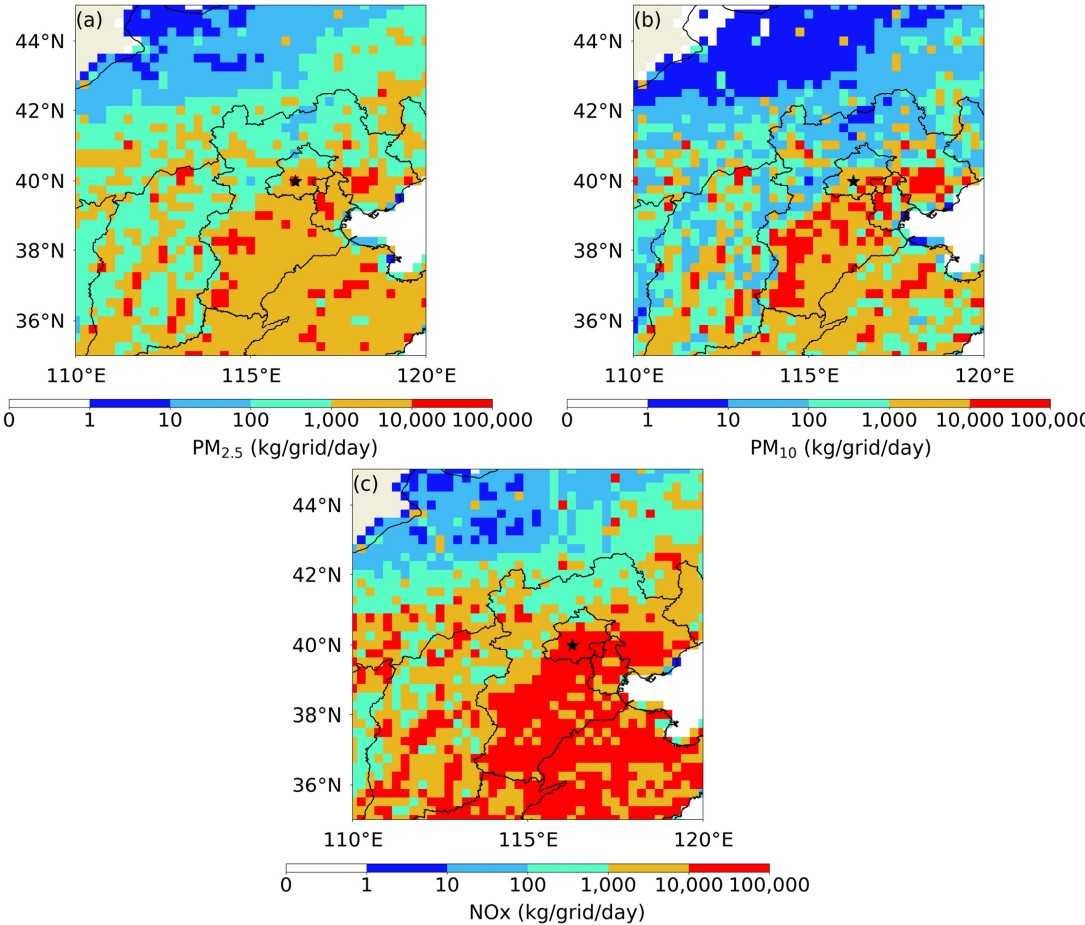

**Figure 8.** Spatial distribution of PM$_{2.5}$, PM$_{10}$, and NOx emissions during spring (March–April–May) 2019, obtained from the multiresolution emission inventory for China (http://meicmodel.org/) (accessed on 21 April 2022). The location of Beijing is marked by the black pentagram.

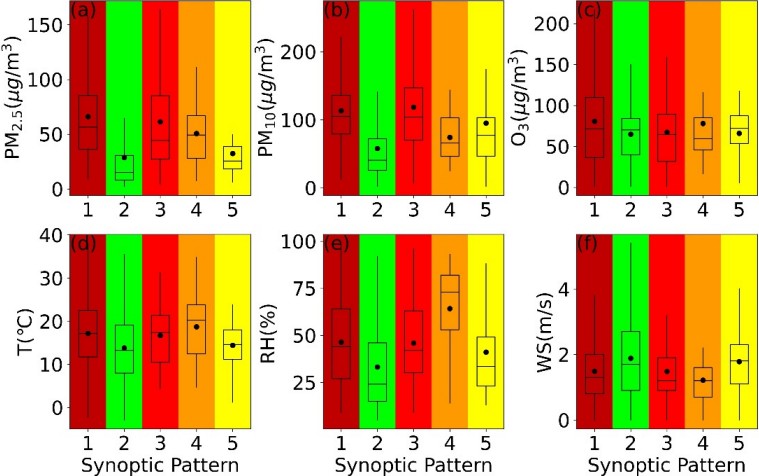

**Figure 9.** Box plots showing the observed (**a**) PM$_{2.5}$ concentration, (**b**) PM$_{10}$ concentration, (**c**) O$_3$ concentration, (**d**) 2-m temperature, (**e**) 2-m relative humidity (RH), and (**f**) 10-m windspeed (WS) related to the different synoptic conditions. The background colour represents the pollution level in Beijing of each synoptic pattern; the pollution becomes severe as the colour gradually changes from green to reddish brown. The symbols in each box are the same as in Figure 2.

## 4. Conclusions

In this study, based on multiple measurements, such as observations from high-resolution RWP and MWR, ERA5 reanalysis data, meteorological observations, and $PM_{2.5}$, $PM_{10}$, and $O_3$ measurements in Beijing in spring 2019, integrated with the T-PCA method and numerical simulations of backward trajectories (HYSPLIT), the variation characteristics of aerosol and ozone pollution under different PBL structures and synoptic patterns in Beijing during spring were comprehensively elucidated.

The BLH retrieved by the RWP during spring in Beijing had an evident diurnal variation, with peaks in the afternoon at 14:00 BJT (average of 1.26 km) and a minimum at night (average of 0.42 km). An anti-correlation existed between the diurnal variation in BLH and $PM_{2.5}$ and $PM_{10}$ concentrations, with correlation coefficients of $-0.12$ and $-0.23$, respectively, which implies that the development of the PBL plays a significant role in modulating aerosol pollution in Beijing. In contrast, diurnal variations in BLH were positively associated with those of $O_3$ concentration (with a correlation coefficient of 0.45), which reached a maximum in the afternoon when solar radiation was strong, promoting photochemical reactions that produce ozone. A similar relationship also existed between the air pollution degree and BH of the TI, and the intensity of the TI was positively associated with $PM_{2.5}$, $PM_{10}$, and $O_3$ concentrations.

Five synoptic flow patterns during spring in Beijing were identified using T-PCA and daily 850-hPa GH fields. Among these patterns, Type 1 featured south-westerly prevailing winds, with high pressure to the south, and was linked with heavy $PM_{2.5}$, $PM_{10}$, and $O_3$ pollution, suggesting that the pollutants emitted from surrounding southern cities were transported to Beijing. In comparison, Type 2, which was driven by north-easterly winds, was related to relatively clean air. In addition to the impact of regional transportation on air quality, high temperature, high RH, and low wind velocity of Type 1 in Beijing favoured the exacerbation of pollution.

This study generally provides significant information on the role of meteorological factors affecting the local PBL structure and their relationship with aerosol and ozone pollution based on observations during spring in Beijing. Although this study focused on the influence of local PBL structures and circulation patterns on pollution, chemical processes cannot be overlooked, and more explicit model simulations (such as WRF–Chem) are warranted in the future.

**Author Contributions:** Investigation, Data curation, Writing—original draft preparation, Q.Z.; Project Administration, Investigation, Conceptualization, L.C.; Conceptualization, Investigation, Writing—review and editing, Y.Z.; Conceptualization, Z.W.; Validation, visualization, S.Y. All authors have read and agreed to the published version of the manuscript.

**Funding:** This research was supported under the auspices of the Ministry of Ecology and Environment of China (DQGG2021101), the National Natural Science Foundation of China (42075044), Key Laboratory of Eco-Environment and Meteorology for The Qinling Mountains and Loess Plateau, Shaanxi Meteorological Bureau (2021K-10), and Shanghai Key Laboratory of Meteorology and Health, Shanghai Meteorological Bureau (QXJK202201).

**Institutional Review Board Statement:** Not applicable.

**Informed Consent Statement:** Not applicable.

**Data Availability Statement:** Not applicable.

**Acknowledgments:** We thank the anonymous reviewers for their constructive suggestions and comments, which helped improve this manuscript.

**Conflicts of Interest:** The authors declare no conflict of interest.

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
