# Peer review of "Relationships between Springtime PM2.5, PM10, and O3 Pollution and the Boundary Layer Structure in Beijing, China"

_sustainability, doi:10.3390/su14159041_

Round 1

Author Response

  First of all, we thank the editor and the reviewers for taking the time to review our manuscript and to offer helpful suggestions. The manuscript has been carefully revised following the suggestions and comments given by the three anonymous reviewers. Please see our detailed point-to-point replies and revised paper in the attachment. 

Reviewer 2 Report

Zhou et al. presented a study using radar wind profilers and microwave radiometer stations, automatic weather stations, and air quality monitoring sites combined with ERA5 and HYSPLIT to understand the relationship between pollutants and boundary layer structure and air mass transport pattern as well as metrological condition in Beijing, China. Overall, this study shows that combining those data can improve our understanding of pollutant transport patterns. The experiment is well designed, the manuscript is concise and clear, and the topic fits well into the scope of the journal. However, I have some comments for the authors. Thus, I suggest the editor consider this manuscript as a revision.

Major comments:

1.       Do you have references for your measuring sites? Also, I want to see more details of your measuring sites, such as altitude, urban or rural, heavy traffic influenced or not, etc. Also, what are the instruments (model, company) you used? What are their sensitivity, resolution, and lower and upper measurement range?

2.       In Section 3.1, I think you are showing very interesting results about the vertical profile of pollutants, which is not well understood. I suggest you add more discussion. I did not really see any discussion of your results. I want to see discussions about what you can tell from your observation. Also, I want to see some discussion of the pollutant source. 

3.       All figures are very blurry. Please use a higher resolution.

Specific comments:

1.       L105-108, “Atmospheric temperature …  inversion algorithms.” Please provide a little bit more details about how to retrieve those data. Did you use any previously developed software or code? Are there any references?

2.       L171, please provide full name, short description, and reference for NCEP-FNL.

3.       Please use the format of hh:mm for time.

4.       L182-185, “However, … reactions.” Do you have any idea why they are positively correlated?

5.       L195-204, “To further … forming ozone,” I do not understand why you use violin plots here. Why don't you use scatter plots of BLH vs. PM2.5, BLH vs. PM10, and BLH vs. O3, which I think will make the correlation more obvious? Why did you only show 8 to 14 BJT? How do you decide the intervals? What are the x and y in your linear regression function? How did you get the function? Moreover, in Figure 3, is there any difference between R and R*?

6.       L235-236, “Based on … (Figure 5).” Please define all 5 patterns.

7.       L240-255, “In these synoptic … pollution level.” What does this tell you? Is this due to transport or local emission? Do you know the local emission factor? Also, please discuss all 5 patterns, not just part of them.

8.       L291-300, “The 24-h … pollution scenario.” For figure 7, are these average backward trajectories or just representative? I suggest showing average with uncertainties as shaded areas. Is figure 8 the average pollutant concentration plot? How much does it vary daily and correlate with daily back trajectories? The pollutant concentration and the pollutant source can be different for the same transport pattern. I suggest at least showing the average value based on the five types of synoptic patterns. Also, please discuss all 5 patterns.

9.       L311-321, “Local meteorological … pollution.” Please discuss all types of patterns

Author Response

  First of all, we thank the editor and the reviewers for taking the time to review our manuscript and to offer helpful suggestions. The manuscript has been carefully revised following the suggestions and comments given by the three anonymous reviewers. Please see our detailed point-to-point replies as the attachment.
